# Nocebo effects are stronger and more persistent than placebo effects in healthy individuals

Angelika Kunkel[1][†], Katharina Schmidt[1][*][†], Helena Hartmann[1], Torben Strietzel[1], Jens-Lennart Sperzel[1], Katja Wiech[1,2][‡], Ulrike Bingel[1][*][‡]

[1]Department of Neurology, Center for Translational Neuro- and Behavioral Sciences (C-TNBS), University Medicine Essen, University Duisburg-Essen, Essen, Germany; [2]Wellcome Centre for Integrative Neuroimaging (WIN), Nuffield Department of Clinical Neurosciences, University of Oxford, John Radcliffe Hospital, Oxford, United Kingdom

*For correspondence:
Katharina.Schmidt@uk-essen.
de (KS);
ulrike.bingel@uk-essen.de (UB)

[†]These authors contributed
equally to this work
[‡]These authors also contributed
equally to this work

Competing interest: The authors
declare that no competing
interests exist.

Reviewing Editor: José Biurrun
Manresa, National Scientific
and Technical Research Council
(CONICET), National University
of Entre Ríos (UNER), Argentina

## eLife Assessment

In this preregistered study, Kunkel and colleagues set out to compare the magnitude and duration of placebo versus nocebo effects in healthy volunteers, and also to examine the different factors contributing to these effects. The authors follow a rigorous methodology in a within-subjects design, taking into consideration standard conventions for manipulation of expectations, and using an appropriate sham condition. They present **compelling** evidence of long-lasting placebo and nocebo effects, with nocebo responses demonstrating consistently greater strength. These **valuable** results have the potential for a great impact in the field of experimental and clinical pain.

**Abstract** Placebo and nocebo effects illustrate the profound influence of cognitive-affective processes on symptom perception and treatment outcomes, with the potential to significantly alter responses to medical interventions. Despite their clinical relevance, the question of how placebo and nocebo effects differ in strength and duration remains largely unexplored. Using a within-subject design in 104 healthy individuals, we investigated and directly compared the magnitude and persistence of placebo and nocebo effects on experimental pain. Effects were assessed immediately after their induction through verbal instructions and conditioning and at a 1-week follow-up. The study was preregistered in the German Clinical Trials Register (registration number: DRKS00029228). Significant placebo and nocebo effects were detected on days 1 and 8, but nocebo effects were stronger on both test days. Sustained effects after 1 week were primarily predicted by individuals' experienced effects on day 1. Our findings underscore the enduring nature of placebo and nocebo effects in pain, with nocebo responses demonstrating consistently greater strength, which is consistent with an evolutionarily advantageous 'better-safe-than-sorry' strategy. These insights emphasise the significant impact of nocebo effects and stress the need to prioritise efforts to mitigate them in clinical practice.

## Introduction

Placebo and nocebo effects are intriguing phenomena that have generated considerable research interest in medicine, psychology, and neuroscience (*Wager and Atlas, 2015*; *Petrie and Rief, 2019*; *Colloca and Barsky, 2020*; *Chen et al., 2024*; *Jensen et al., 2015*). Belief in the effectiveness or ineffectiveness of a treatment can reduce or increase symptoms, highlighting the powerful interaction

between perception, physiology and cognitive-affective processes. Harnessing the power of positive expectations could complement standard medical treatments, and thereby enhance overall treatment outcome (*Enck et al., 2013*; *Bingel, 2020*). Conversely, awareness of nocebo effects is important to minimise negative expectations and side effects in clinical practice (*Colloca and Barsky, 2020*; *Bingel and Placebo Competence Team, 2014*). Moreover, it is relevant in placebo-controlled clinical trials where nocebo effects, manifesting as adverse events in the placebo group, can decrease treatment adherence and even lead to treatment discontinuation (*Colloca, 2024*). Recent insights into both phenomena have therefore led to a growing call to systematically utilise placebo effects and to learn to avoid nocebo effects in clinical care.

While extensive investigations have focused on the psychological and neurobiological mechanisms underlying positive expectations and their effect on symptom perception (*Petrie and Rief, 2019*; *Bingel, 2020*), our understanding of negative expectations and nocebo effects is comparably sparse despite evidence that nocebo effects can be moderate to large in size (*Petersen et al., 2014*). Even less is known about the longevity of the effect, a crucial factor for assessing its impact on treatment outcome in real-life scenarios.

Importantly, there is evidence suggesting that an individual's susceptibility to nocebo information may not simply mirror their capacity for placebo analgesia. Early research by *Colloca et al., 2010* demonstrated that a single session using non-painful stimuli induced a nocebo effect but failed to elicit a placebo effect, indicating that negative expectations may be more readily triggered than positive ones. Moreover, nocebo effects seem to generalise more easily to other symptoms or treatments (*Zunhammer et al., 2017*; *Faasse et al., 2019*). Given the evolutionary relevance of anticipating negative, threatening, and potentially harmful events, it seems reasonable to assume that negative expectation and its effect on health outcome is an integral aspect of promoting safety behaviours and is thus more persistent than positive expectation. To accurately gauge an individual's capacity to produce placebo and nocebo effects and compare their magnitude and duration, it is essential to investigate both effects within the same individual.

Here we investigated immediate and sustained effects of positive and negative treatment expectations on experimentally induced heat pain in N = 104 healthy volunteers. Our experimental approach allowed for the trial-by-trial modulation of expectations for pain relief and pain aggravation in a within-subject design. Verbal instructions were combined with conditioning along with a sham electrical stimulation, which was introduced to participants as a method to 'induce frequency-dependent changes in pain sensitivity'. Treatment expectations and pain perception of physically identical medium-level heat stimuli were assessed immediately after expectancy induction (day 1), but also 1 week later (day 8) to investigate the longevity of both placebo analgesia and nocebo hyperalgesia. We also assessed psychological variables to explore whether they modulate or predict an individual's susceptibility, effects, and persistence of expectancy effects on pain. We hypothesised that negative expectations and nocebo effects would be stronger than positive expectations and placebo effects induced on day 1, and that negative expectations and their effects are more resistant to extinction and would therefore still be stronger on day 8.

Our data confirm that, although significant placebo and nocebo effects were found on days 1 and 8, the nocebo effect was consistently stronger. Both effects were primarily influenced by the most recent experience of pain reduction and pain increase but were also susceptible to psychological factors.

## Results

The calibration procedure determined one temperature level for the placebo condition and one for the nocebo condition that were equidistant from the temperature of the control condition. These three temperatures were used in the conditioning procedure to induce the perception of pain reduction (placebo hyperalgesia) and pain aggravation (nocebo hyperalgesia), respectively (for details, see *Figure 1* and Appendix 1). In the test sessions on days 1 and 8, however, the same medium-level temperature of the control condition was applied in all three conditions. The analyses include comparisons between all three conditions (i.e. placebo, nocebo and control) and comparisons between placebo effects (i.e. control vs. placebo) and nocebo effects (i.e. nocebo vs. control). To identify variables associated with placebo or nocebo effects on day 1 or day 8, we conducted multiple regression

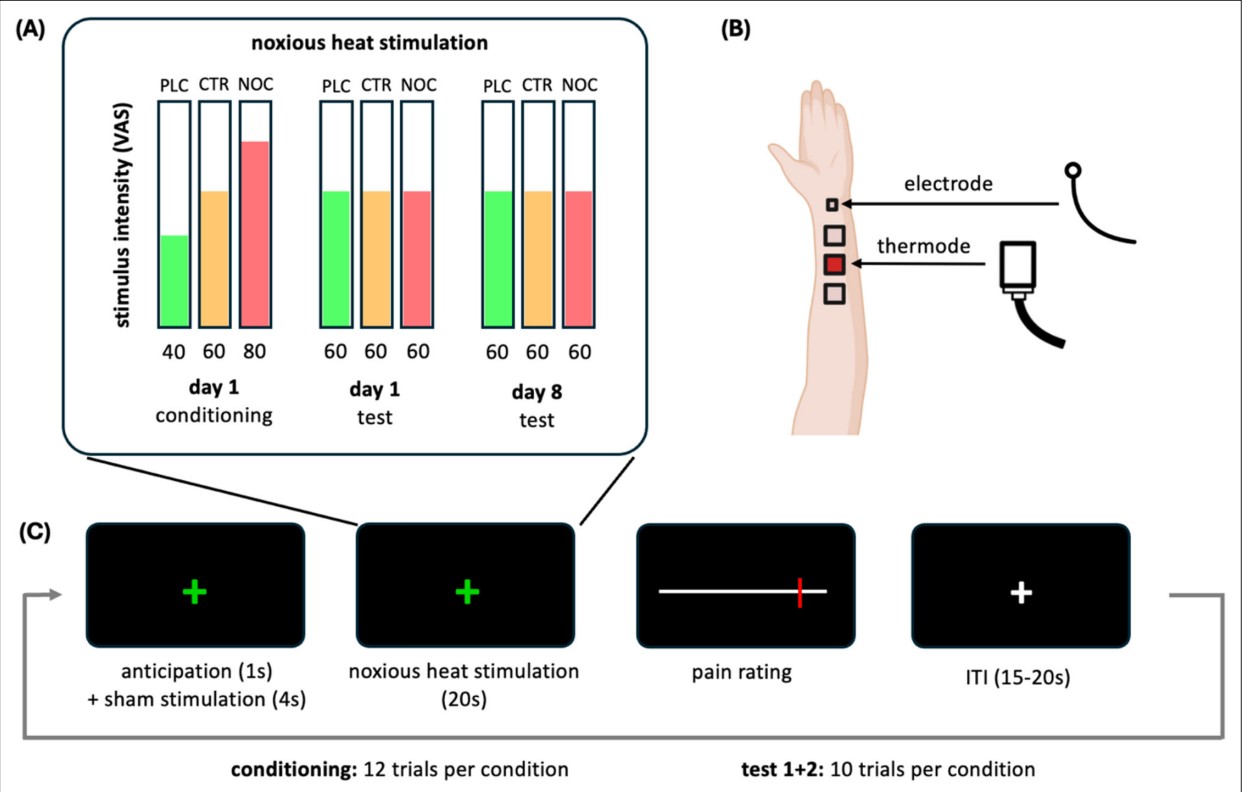

**Figure 1.** Study and trial design. (**A**) Study design: on day 1, participants underwent a conditioning procedure in which a noxious heat was applied directly after a (sham) TENS: (transcutaneous electric nerve stimulation) stimulation in three conditions. In the placebo condition (PLC), the thermal stimulation was lowered to VAS 40, in the nocebo condition (NOC), it was increased to VAS 80 and in the control condition (CTR) it remained unchanged (VAS 60). During the two tests on days 1 and 8, the same moderate stimulation intensity of VAS 60 was applied in all three conditions. (**B**) Position of the electrode on the inner lower left arm for (sham) TENS stimulation (approximately 2.5 cm above the wrist) and the thermode at three possible locations (approximately 3.5 cm above the electrode with a distance of 0.5 cm between each of the three locations). (**C**) Trial design: following the presentation of a visual cue to indicate the condition (e.g. green cross for the placebo condition), first the sham TENS stimulation and then the heat stimulus were applied before participants rated the pain intensity on a 0–100 visual analogue scale.

## Placebo and nocebo effects on day 1

The comparison of pain intensity ratings acquired after the conditioned expectancy manipulation in the first test session on day 1 confirmed differences between three conditions ($F$(1.28, 131.96) = 96.32, $P<0.001$) with both a significant placebo effect (control vs. placebo condition: $t$(103) = 3.92; $P<0.001$; 95% CI, 2.07–6.32; $d = 0.38$) and a significant nocebo effect (nocebo vs. control condition: $t$(103) = 14.88; $P<0.001$; 95% CI, 9.78–12.79; $d = 1.46$; *Figure 2A*). A direct comparison of both effects revealed a stronger nocebo effect than placebo effect (nocebo effect: M = 11.29, SD = 7.73; placebo effect: M = 4.19, SD = 10.92; $t$(103) = 6.56; $P<0.001$; 95% CI, 4.95–9.24; $d = 0.64$; *Figure 2B*).

## Placebo and nocebo effects on day 8

In the second test session, seven days after the expectancy manipulation, pain intensity ratings remained to be different between conditions ($F$(1.58, 153.34) = 111.93, $P<0.001$), despite the same stimulation intensity. Participants still showed a significant placebo effect (control vs. placebo condition: $t$(97) = 6.06; $P<0.001$; 95% CI, 3.08–6.09; $d = 0.61$) as well as a nocebo effect ($t$(97) = 10.79, $P<0.001$; 95% CI, 7.29–10.58; $d = 1.09$). As on day 1, a direct comparison between both effects using difference scores showed a stronger nocebo than placebo effect on day 8 (nocebo effect: M = 8.93, SD = 8.20; placebo effect: M = 4.58, SD = 7.50; $t$(97) = 3.90, $P<0.001$, 95% CI, 2.14–6.56; $d = 0.39$) (*Figure 2B*).

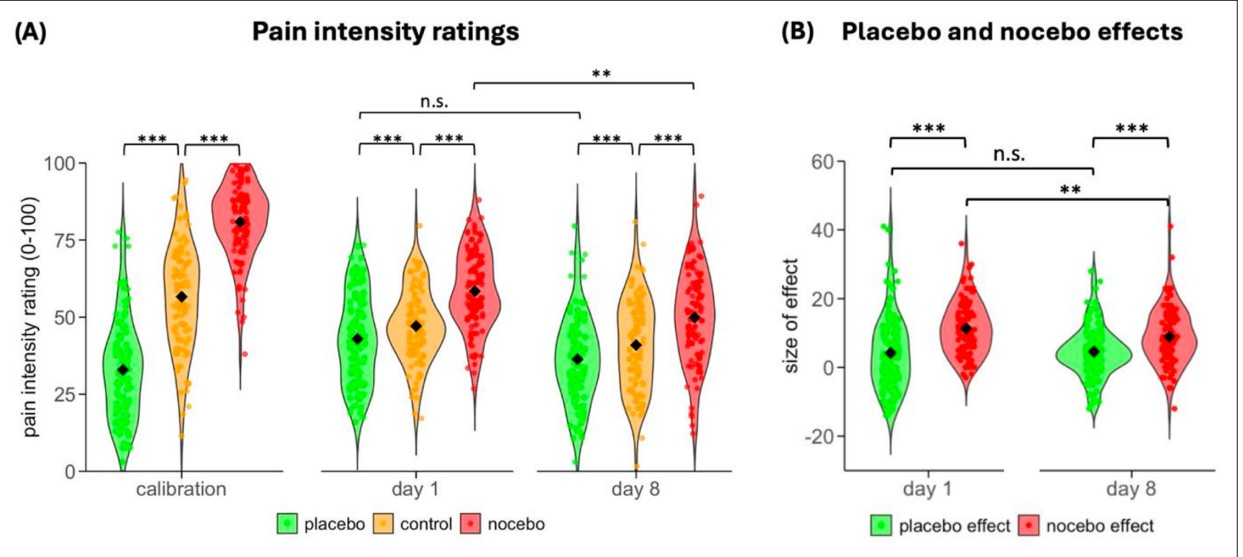

**Figure 2.** Pain intensity ratings and placebo and nocebo effects during calibration and test sessions. (**A**) Mean pain intensity ratings in the placebo, nocebo, and control condition during calibration, and during the test sessions at days 1 and 8. (**B**) Placebo effect (control condition – placebo condition, i.e. positive value of difference) and nocebo effect (nocebo condition – control condition, i.e. positive value of difference) on days 1 and 8. Black diamond shapes indicate the mean and circles the individual scores. \*\*\*P<0.001, \*\*P<0.01, n.s.: non-significant.

## Comparison of days 1 and 8

A direct comparison of placebo and nocebo effects on day 1 and 8 showed a main effect of *Effect* with a stronger nocebo effect ($F_{(1,97)}$ = 53.93, $P$<0.001, $\eta^2$ = 0.36) but no main effect of *Session* ($F_{(1,97)}$ = 2.94, $P$ = 0.089, $\eta^2$ = 0.029). The significant *Effect × Session* interaction indicated that the placebo effect and the nocebo effect developed differently over time ($F_{(1,97)}$ = 3.98, $P$ = 0.049, $\eta^2$ = 0.039). While the nocebo effect decreased significantly from day 1 to day 8 ($t_{(97)}$ = 2.68, $P$ = 0.018, 95% CI, 0.66–4.44; $d$ = 0.27), the placebo effect did not change ($t_{(97)}$ = –0.517; $P$ = 0.606; 95% CI, –2.47–1.45, $d$ = –0.05), possibly due to the lower starting point on day 1. Of note, placebo and nocebo effects were significantly positively correlated at day 1 ($r$ = 0.34; $P$<0.001) but showed no significant relationship on day 8 ($r$ = 0.01; $P$ = 0.903).

## Evolution of differences between placebo and nocebo effects

To test whether the difference between the placebo and the nocebo condition already evolved during conditioning, we first compared pain intensity ratings provided during conditioning where stimulus

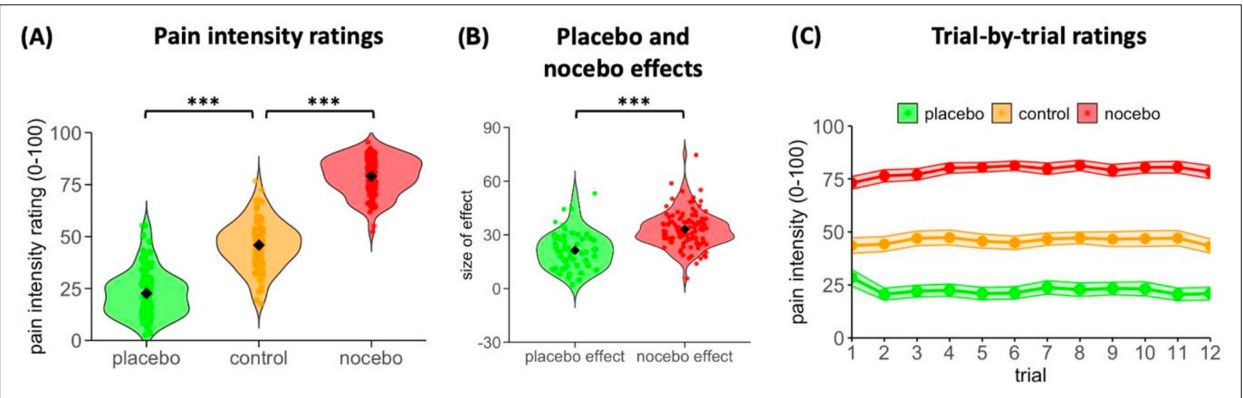

**Figure 3.** Mean and trial-by-trial pain intensity ratings, placebo and nocebo effects during conditioning. (**A**) Mean pain intensity ratings of the placebo, nocebo and control condition during conditioning. (**B**) Placebo effect (control condition – placebo condition, i.e. positive value of difference) and nocebo effect (nocebo condition – control condition, i.e. positive value of difference) during conditioning. (**C**) Trial-by-trial pain intensity ratings (with confidence intervals) during conditioning. Black diamond shapes indicate the mean and circles the individual scores. \*\*\*P<0.001.

intensities were manipulated unbeknownst to the participant. As intended, heat stimuli applied during placebo conditioning were rated as less painful than stimuli applied in the control condition (control vs. placebo condition: $t(103) = 20.56$; $P<0.001$; 95% CI, 20.98–25.45; $d = 2.02$). Similarly, stimuli applied during nocebo conditioning were rated as more intense than stimuli in the control condition: $t(103) = 33.42$; $P<0.001$; 95% CI, 31.16–35.09; $d = 3.28$ (*Figure 3A*). However, the pain ratings revealed a stronger conditioning effect for the nocebo condition than the placebo condition (pain worsening effect: M = 33.12, SD = 10.11, pain relief effect: M = 23.21, SD = 11.51; $t(103) = 5.96$, $P<0.001$, 95% CI, 6.61–13.20; $d = 0.59$, *Figure 3B*).

To explore the formation of the pain relief and pain worsening during conditioning in more detail, we compared changes in trial-by-trial pain intensity ratings over the conditioning phase between the three conditions (*Figure 3C*). This analysis showed no significant main effect of *Trial* ($F(4.37,341.01) = 1.25$, $P = 0.289$, $\eta^2 = 0.016$), indicating that there was no general change in ratings over time. However, as shown by a significant main effect of *Condition* ($F(1.84,143.76) = 950.85$, $P<0.001$, $\eta^2 = 0.924$) and more importantly a significant interaction between *Trial* and *Condition* ($F(13.93,1086.45) = 4.93$, $P<0.001$, $\eta^2 = 0.059$), changes in ratings over time differed between the three conditions. Separate regres-

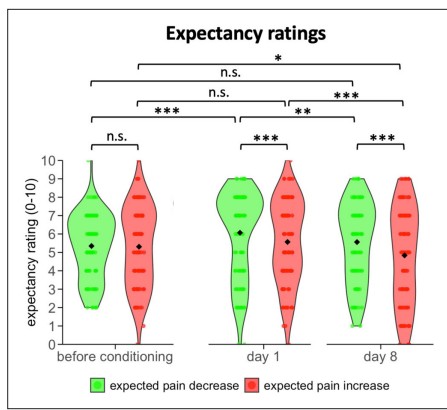

**Figure 4.** Expectancy ratings obtained before conditioning and before the test sessions on days 1 and 8. Expectations were assessed using the *Generic Rating Scale for Previous Treatment Experiences, Treatment Expectations, and Treatment Effects* (*Rief et al., 2021*). The expected pain relief was derived from the item asking how much improvement the participant expected from the treatment on a 10-point Likert scale from 0 (=no improvement) to 10 (=greatest improvement imaginable). Analogously, the expected pain increase (nocebo effect) was taken from the item asking how much worsening of pain they expected from the treatment from 0 (=no worsening) to 10 (=greatest worsening imaginable). Black diamond shapes indicate the mean and circles the individual scores. ***$P<0.001$, **$P<0.01$, *$P<0.05$, n.s.: non-significant.

sion analyses for each condition showed that, although ratings decreased in the placebo condition ($\beta = -0.22$), the decrease was not significant ($P = 0.242$). Conversely, both the nocebo and the control condition showed an increase in ratings over time, but the increase only reached significance in the nocebo condition ($\beta = 0.39$, $P = 0.048$; control condition: $\beta = 0.09$, $P = 0.512$), indicating a stronger formation of nocebo hyperalgesia already during conditioning, despite rigorous calibration to intensities equidistant from the control condition.

To test whether the differences between placebo effects and nocebo effects on days 1 and 8 could be explained by stronger nocebo conditioning, we repeated the previous comparisons between both effects, but this time included the difference in conditioning (nocebo condition - placebo condition) as a covariate. While the difference in conditioning could indeed explain a significant part of the variance ($F(1,102) = 5.85$, $P = 0.017$, $\eta^2 = 0.054$), the nocebo effect was still significantly stronger on day 1 (main effect *Effect*: $F(1,102) = 20.79$, $P<0.001$, $\eta^2 = 0.169$), indicating genuine differences in the underlying mechanisms and temporal dynamics. A similar (albeit weaker) result was found for day 8 with a significant difference between the placebo and the nocebo effect (main effect *Effect*: $F(1,96) = 4.81$, $P = 0.031$, $\eta^2 = 0.048$) in addition to a significant effect of the difference in conditioning ($F(1,96) = 4.38$, $P = 0.039$, $\eta^2 = 0.044$).

## Expectancy ratings

Given the proposed key role of expectations in placebo and nocebo effects, we also obtained expectancy ratings prior to each testing session. Because expectancy ratings were not normally distributed, we used a non-parametric analysis approach. Expectations that the pain would improve in the placebo condition and worsen in the nocebo condition did not differ significantly before conditioning, confirming that our verbal instruction had induced equally strong expectations ($Z(104) = -0.34$, $P = 0.737$; *Figure 4*). The conditioning procedure on day 1 significantly increased the expected pain relief

(placebo) ($Z$(104) = –3.76, $P$<0.001) but not the expected pain worsening (nocebo) ($Z$(104) = –1.09, $P$ = 0.556) and a direct comparison showed significantly stronger placebo than nocebo expectations ($Z$(104) = –2.71, $P$ = 0.007). Between days 1 and 8, placebo expectations decreased significantly ($Z$(98) = –3.09, $P$ = 0.004) and were no longer different from ratings before conditioning ($Z$(104) = –0.96, $P$ = 0.338). Nocebo expectations also decreased ($Z$(98) = –3.90, $P$<0.001) and were even significantly lower than before conditioning ($Z$(98) = –2.30, $P$ = 0.021). As on day 1, the expected pain relief was significantly stronger than the expected worsening of pain ($Z$(98) = –3.39, $P$ = 0.001).

Neither placebo nor nocebo expectations were significantly linked to the experienced effect on day 1 (placebo: Spearman's rho (104) = 0.10, $P$ = 0.335; nocebo: Spearman's rho (104) = 0.17, $P$ = 0.093) or day 8 (placebo: Spearman's rho (98) = 0.13, $P$ = 0.187; nocebo: Spearman's rho (98) = 0.88, $P$ = 0.396).

## Multiple linear regression analyses (expected and experienced effects)

Next, we employed multiple linear regression analyses to investigate the significance of expected (GEEE ratings [Generic rating scale for previous treatment experiences, treatment expectations, and treatment effects]) and experienced placebo and nocebo effects (visual analogue scale [VAS] ratings) for subsequent effects on both test days. Overall, the regression model for the placebo effect on day 1 explained 9.7% of the variance (*Appendix 1—table 1*). The only predictive variable for the placebo response on day 1 was the pain relief during conditioning. In the equivalent model for the nocebo effect, none of the variables could significantly predict the nocebo response on day 1.

The regression model for the placebo effect on day 8 explained a total of 25.1% of the variance with two significant predictors: the placebo effect on day 1 and the placebo expectation on day 8. For the nocebo response on day 8, the tested model explained 7.1% of the variance with the nocebo effect at day 1 as the only significant predictor (*Appendix 1—table 1*). Together, these differences in the contribution of expectations and experienced effects between the placebo and the nocebo condition further substantiate that both effects are driven by different mechanisms.

## Multiple regression analyses (expected and experienced effects plus psychological variables)

In the final analysis step, we tested whether psychological variables that have been linked to placebo and nocebo effects in the past, such as trait anxiety (*Kern et al., 2020*), practitioner characteristics (*Howe et al., 2017*), or somatosensory amplification (*Doering et al., 2015*) could increase the predictive power of the previously tested models. On day 1, in addition to the significant prediction from the experienced conditioning effect that had already been significant in the previous model, somatosensory amplification emerged as a negative predictor of the placebo effect, indicating that individuals with a higher tendency for somatosensory amplification were less likely to experience placebo analgesia. The total variance explained in this model was 14.5% (*Appendix 1—table 2*). This influence of somatosensory amplification was no longer detectable on day 8 where only the experienced placebo effect on day 1 and placebo expectations on day 8 were significant predictors but none of the psychological variables (total amount of variance explained: 26.4%).

The equivalent analyses for the nocebo effect revealed that higher nocebo effects were found when participants had rated the experimenter competence as high (*Appendix 1—table 2*), pointing towards a potential iatrogenic effect of experimenter when they implied that pain could become worse with the treatment. The total amount of variance explained by this model was 10.6%. As for the placebo effect, none of the psychological variables predicted the nocebo effect on day 8. The total variance explained by this model with only the perceived nocebo effect on day 1 as a significant predictor was 1.6%.

## Discussion

In this preregistered, experimental study in healthy individuals, we investigated placebo analgesic and nocebo hyperalgesic effects immediately after a conditioned expectancy manipulation and seven days later. Three key findings emerged from our investigation. First, medium-to-large scale placebo and nocebo effects were found not only on day 1 but also 1 week later. Second, nocebo effects were consistently stronger than placebo effects, including during the conditioning phase, despite

analogous conditioning protocols in both conditions. Third, placebo and nocebo effects are primarily driven by the most recent experience of these effects but were also susceptible to some psychological factors.

## Sustained placebo and nocebo effects

While placebo effects have been shown to persist for an extended period of time after they have been induced, there are only a few studies that have investigated the longevity of nocebo effects so far and these studies focused on sustained effects within the same test session (*Colloca et al., 2010*; *Colagiuri et al., 2015*). In our study, nocebo effects were not only sustained over the period of a week, but they were also significantly stronger than the placebo effect on both test days (*Figure 2B*). This finding aligns with broader evidence from learning studies, which demonstrate a greater influence of negative information on sensory perception (*van der Schaaf et al., 2022*; *Forkmann et al., 2023*; *Zika et al., 2023*), as well as similar effects observed in placebo and nocebo trials. For example, nocebo hyperalgesia was more easily induced via instructions than placebo analgesia (*Colloca et al., 2008*) and tended to extinguish more slowly (*Colagiuri et al., 2015*; *Colloca et al., 2008*). Additionally, in an experimental study involving healthy individuals, *Colloca et al., 2010* found that one session of conditioning was sufficient to induce a nocebo effect but not a placebo effect.

Stronger and more sustained nocebo effects are likely to be the result of a combination of different factors. Evolutionary psychology suggests that humans may have evolved to be more attuned to potential threats for survival. Negative information or expectations about harm may have carried more evolutionary significance, making individuals more sensitive to nocebo suggestions, a tendency often referred to as 'better safe than sorry'. Confirmation for this assumption comes from brain imaging studies demonstrating a cognitive bias in which the brain tends to process negative information more readily than positive information. Moreover, negative expectations and fear tend to amplify sensory perception (*van der Schaaf et al., 2022*; *Forkmann et al., 2023*; *Zika et al., 2023*). When individuals anticipate a negative outcome, their attention is often heightened which makes them susceptible to perceiving symptoms, even in the absence of an actual stimulus. In line with this assumption, nocebo effects have been shown to lead to anticipatory anxiety and autonomic arousal which mediated the effect on extinction in an experimental learning model (*Colagiuri et al., 2015*). It may be argued that the dominant nocebo effect observed in our study is the result of the stronger conditioning in the nocebo condition (*Figure 3*). This asymmetry is noteworthy in and of itself because it occurred despite the equidistant stimulus calibration relative to the control condition prior to conditioning. It may be the result of different physiological effects of the stimuli over time or amplified learning in the nocebo condition, consistent with its heightened biological relevance, but it could also be a stronger effect of the verbal instructions in this condition. Importantly, the stronger nocebo effect observed on both test days remained significant even after accounting for the asymmetric conditioning effect, ruling out that conditioning differences alone explain the stronger nocebo effects. Instead, it suggests that the two effects may be induced and maintained by at least in part distinct mechanisms and temporal dynamics. Recent work using a predictive coding framework further suggests that nocebo effects may be less susceptible to prediction error than placebo effects (*Hird et al., 2019*), which could contribute to their greater persistence and strength in our study. This is supported by the observation that, similarly to a previous study (*Colloca et al., 2010*), a significant correlation between placebo and nocebo effects was found on day 1 but was no longer detectable at the follow-up 1 week later. Interestingly, our expectancy manipulation increased placebo expectations, but had no significant effect on nocebo expectations (*Figure 4*). Furthermore, expectations were not correlated with actual placebo or nocebo effects on either test day. While this may seem surprising, it has recently been suggested that these correlations depend on whether expectations are measured in the same format as the pain experience or as a difference measure, as in our study (*Lunde et al., 2024*). Further research is therefore needed to investigate the effects of assessment methods on such associations.

It is important to note that our study was designed in alignment with previous studies addressing similar questions (e.g., *Colloca et al., 2010*). Our primary aim was to directly compare placebo and nocebo effects in a within-subject design and assess their persistence of these effects 1 week following the first test session. One limitation of our approach is the relatively short duration of each session, which may have limited our ability to examine the trajectory of responses within a single session.

Future studies could address this limitation by increasing the number of trials for a more comprehensive analysis.

## Past effects predict future effects

To explore the relative influence of expectations and prior experience in more detail, we conducted separate regression analyses for placebo and nocebo effects on both test days, using expectations and perceived effects as predictors. The analyses revealed that experienced pain reduction and increase were significant predictors of subsequent effects, especially for the placebo effect on days 1 and 8, and for the nocebo effect on day 8 (*Appendix 1—table 1*). This highlights the strong impact of sensory experience on subsequent effects, in line with studies on learning (*Atlas et al., 2016*), meta-analyses of behavioural placebo analgesia (*Vase et al., 2002*), and previous studies on carry-over effects between analgesic treatments (*Zunhammer et al., 2017*). Notably, the most recent experience was the most predictive in all three analyses; for instance, the placebo effect on day 8 was predicted by the placebo effect on day 1, not by the initial conditioning. This finding supports the Bayesian inference framework, where recent experiences are weighted more heavily in the process of model updating because they are more likely to reflect the current state of the environment, providing the most relevant and immediate information needed to guide future actions and predictions (*Büchel et al., 2014*). Interestingly, while a change in pain predicted subsequent nocebo effects, it seemed less influential than for placebo effects. This aligns with findings that longer conditioning enhanced placebo effects, while it did not affect nocebo responses (*Colloca et al., 2010*) and the conclusion that nocebo instruction may be sufficient to trigger nocebo responses. Using Bayesian modelling, future studies could identify individual differences in the development of placebo and nocebo effects by integrating prior experiences and sensory inputs, providing a probabilistic framework for understanding the underlying mechanisms.

## The role of psychological variables in immediate and sustained placebo and nocebo effects

Our extended regression models, incorporating psychological variables, highlight two interesting predictors: somatosensory amplification and perceived practitioner competence (*Appendix 1—table 2*). Somatosensory amplification, described as a tendency to experience bodily symptoms as intense, noxious and disturbing (*Doering et al., 2015*), was associated with a weaker placebo effect on day 1. This may be due to higher-level evaluative processes (*Nakao et al., 2007*), leading individuals to perceive symptoms as more threatening, which in turn diminishes the influence of cognitive processes that typically drive placebo effects. Additionally, our study suggests that nocebo effects can be linked to the perceived competence of the experimenter. While practitioner competence – alongside perceived warmth – usually enhances positive treatment expectations (*Seewald and Rief, 2024*) and treatment outcome (*Howe et al., 2017*; *Ashton-James et al., 2019*), it might also make negative suggestions more convincing and thereby amplify nocebo responses through increased anxiety or hypervigilance. This finding underscores the dual-edged nature of competence in patient–practitioner interactions, where heightened credibility could inadvertently strengthen nocebo effects.

Our findings have important implications for clinical research and practice. First, they underscore the necessity of prolonged observation periods in clinical trials to accurately capture the durability of these effects. Second, they emphasise the importance of not dismissing early signs of nocebo effects as they may persist and undermine otherwise treatments if left unaddressed. Third, our findings advocate for a stronger focus on the prevention of nocebo effects. While considerable effort has been made to leverage placebo effects, it is equally – if not more – crucial to focus on minimising nocebo effects, which seem to be triggered more easily. Fortunately, nocebo effects can often be avoided by adopting simple, effective strategies to improve patient–practitioner communication. For example, positive framing, avoiding unnecessary focus on potential side effects or building a trusting relationship can reduce the likelihood of triggering nocebo effects. In a time where cost-effectiveness is paramount, and healthcare resources must be carefully allocated, prioritising the prevention of nocebo effects should be a key strategy to enhance treatment outcome and reduce overall healthcare costs.

In summary, our findings indicate that nocebo effects are indeed more than the flipside of a placebo effect and that the two phenomena may be sustained by distinct mechanisms. These insights

shed light on the factors that exacerbate nocebo effects and underscore the importance of carefully managing communication in clinical and experimental settings.

## Materials and methods

### Participants

As there are no studies investigating within-subject effects that could provide indications for the relevant effect sizes, the study was powered to detect small effect sizes for the hypothesised differences between the placebo and nocebo condition ($d$ = 0.2–0.25, $\alpha$ = 0.05, power = 0.95, $N$ = 84–120). A total of $N$ = 112 healthy volunteers were recruited through public adverts and received structured telephone interviews for screening purposes. Exclusion criteria comprised red-green colour blindness, drug use in the last four weeks, alcohol consumption in the last 24 hours, caffeine consumption on the test day, acute or chronic pain, a history of or acute psychiatric disorders (including major depression, schizophrenia, and suicidality), hypersensitivity or other neurological diseases, acute infections, skin diseases, surgical procedure under anaesthesia in the last six months, use of analgesic or anticoagulant medications within the last 24 hours, use of any other medication in the last seven days (except thyroid medication, hormonal contraceptives, or allergy medication), pregnancy, or breastfeeding. People were also ineligible if they had taken part in another study using electrical stimulation or experimental heat pain in the last six months before the study. Eight participants were excluded on the first testing day, two because of technical problems, two because they did not meet the inclusion criteria (due to caffeine consumption and yellow fever vaccination), and four showed a low or inconsistent pain sensitivity rendering the experimental manipulation ineffective (e.g., 80% of the pain stimuli were rated with a VAS score of zero). The final sample for the analyses of day 1 consisted of 104 participants (63 females and 41 males, mean ± SD age: 24.92±3.47, range = 18–36 years). Six participants were unable to take part in the follow-up examination on day 8 for the following reasons: one due to personal illness, two because of the experimenter's illness, one failed to attend, another participated in a similar experiment between sessions, and one took pain medication on day 8. As a result, the final sample for day 8 consisted of 98 participants (59 females, 39 males, mean age ± SD: 24.86±3.29 years, range: 18–36 years). The study was preregistered with the German Clinical Trials Register (https://drks.de/search/de/trial/DRKS00029228; registration number: DRKS00029228). Ethics approval was granted by the University Hospital Essen (22-10597-BO). The experiment adhered to the principles outlined in the 2013 Declaration of Helsinki. Informed written consent was obtained from all participants, who received 120 euros for their participation.

### Study design and procedure

This study used a within-subjects design (*Figure 1*) to investigate the immediate and sustained effects of three types of experimentally induced treatment expectations on heat pain perception: expectations of reduced pain (placebo condition), expectations of increased pain (nocebo condition), and expectations of no change in pain (control condition). The experiment was carried out on 2 days with an approximate duration of 3 hours on day 1 and 1 hour on day 8. On the first day (day 1), treatment expectations were induced using verbal instructions in combination with a conditioning procedure. During conditioning, participants learned to associate the presentation of one of three visual, differently coloured cues with a reduction of heat-induced pain through a (sham) 'transcutaneous electrical nerve stimulation (TENS) device' that was introduced as an analgesic treatment in the placebo condition. A second cue signalled an increase in pain in the nocebo condition, and the third cue signalled no change in pain in the control condition. As in previous studies using conditioning to induce placebo and nocebo effects (*Colloca et al., 2010*; *Colloca and Benedetti, 2006*; *Montgomery and Kirsch, 1997*; *Voudouris et al., 1990*), unbeknownst to the participant, the heat stimulation was reduced from VAS 60 to VAS 40 in the placebo condition, increased to VAS 80 in the nocebo condition and left unchanged at VAS 60 in the control condition. In the subsequent first test session, the same moderate stimulation intensity of VAS 60 was used in all three conditions. To explore the longevity of the induced conditioned effects, participants underwent the same testing procedure but no conditioning a week later (day 8) with all three visual stimuli again followed by the same moderate temperature stimulation (VAS 60). Participants' condition-specific treatment expectations and trial-by-trial pain intensity ratings were recorded as outcome measures. The study also comprised structural

and functional MRI that took place on a separate day before day 1 (methods and data on this part will be reported elsewhere).

During the experiment, the participants were seated in a chair in front of a computer in a behavioural laboratory setting with a keyboard as response device. The left arm was positioned on a long cushion resting on the table while the right hand operated the keyboard. The experimenter faced the participant from the opposite side of the table with the computer screen between them.

Presentation of visual stimuli, delivery of thermal and electrical stimuli and outcome recording were implemented using Presentation (version 22.0, Neurobehavioral Systems, Inc, Berkeley, CA).

## (Sham) transcutaneous electrical nerve stimulation (TENS)

Participants were instructed that the applied non-painful electrical stimulation with different frequencies would either increase, decrease, or not influence pain perception, respectively. The electrical stimuli were applied to the left volar forearm approximately 2.5 cm proximal of the wrist using a Digitimer stimulator (Welwyn Garden City, England, model DS7A) that was connected to a surface electrode (Specialty Developments, Bexley, UK) with a diameter of approximately 5 mm attached to the skin using medical tape. During calibration, the initial stimulation intensity for 500 ms stimuli started at 0.9 mA and increased in increments of 0.1 mA until participants noticed a clear but non-painful sensation. This intensity was then tested by applying four 4-second stimuli. If participants rated at least two out of four of the stimuli between 25 and 35 on a VAS from 0 to 100 (anchors: 0 = not perceivable, 100 = unbearably painful; 25-35 equals perceivable but not painful), this final stimulation intensity was carried forward to be used throughout the test sessions. If the electrostimulation was not perceivable on day 8, the calibration was repeated once more before the start of the other experiments.

## Calibration of the noxious thermal stimulation

Heat stimuli were calibrated to each participant's level of sensitivity. First, we used the Method of Limits (*Fruhstorfer et al., 1976*) to determine the individual heat pain threshold (HPT) in three consecutive trials. In the subsequent calibration procedure, participants rated 21 noxious heat stimuli with varying temperature levels around the HPT (–1°C to 3.5°C) on a VAS with endpoints 0 (='not painful at all') and 100 (='unbearably painful'). These ratings were entered into a linear regression (lm(VAS rating ~temperature)) in *R Development Core Team, 2021* (except for the first rating due to familiarisation effects) to determine the temperature levels rated as VAS 40, 60, and 80. These temperatures were applied twice in a short subsequent test to ensure that the calculated heat levels induced the intended pain intensity. The 20-second contact heat pain stimuli were delivered using a Pathway advanced thermal stimulator with a 30 × 30 mm activation area (Pathway System, Medoc, Israel). The thermode was attached to one of three possible locations on the medial inner aspect of the left forearm using a tourniquet, maintaining a standardised distance of 3.5 cm from the electrode maintained via a template. To prevent sensitisation or habituation, three different stimulation sites were used. The thermode was moved to another of the three locations after calibration and conditioning, following a pseudorandomised order.

## Conditioning procedure

During the conditioning session, participants' expectations of pain relief and pain increase were modulated using verbal instructions and electrical stimulation coupled with coloured visual cues. Specifically, participants were told that the electrical stimulation would either increase pain (nocebo instruction), decrease pain (placebo instruction), or have no influence on their pain perception (control instruction) depending on the frequency of the stimulation. The direction of change would be indicated by the colour of a cross that was shown in the centre of the computer screen. A green cross indicated a decrease in pain (placebo condition), a red cross indicated an increase of pain (nocebo condition) and a yellow indicated no change (control condition). In fact, unbeknownst to the participants, in placebo trials the green cross was followed by low-intensity heat stimulation calibrated at VAS 40 to induce a sense of pain reduction through the electrical stimulation, whereas the red cross was followed by a high-intensity heat pain calibrated at VAS 80 for a sense of pain increase (*Figure 1*). In control trials, the yellow cross was followed by a VAS 60 heat pain stimulus. This temperature manipulation was applied to all trials of the conditioning procedure, respectively. The order of conditions was pseudorandomised, and each trial type was repeated twelve times during the conditioning

procedure. Due to a randomisation error, 25 participants received an unbalanced number of trials per condition (i.e. 10x VAS 40, 14x VAS 60, 12x VAS 80). However, mean pain intensity ratings during the conditioning phase did not differ significantly between these participants and the remaining sample in any of the three conditions (two-sample $t$-test (two-sided); placebo condition: $t(102) = -0.806$, $P = 0.422$), nocebo condition: ($t(102) = 0.849$, $P = 0.398$), control condition: ($t(102) = 0.390$, $P = 0.697$).

### Test sessions

Placebo and nocebo responses were assessed during both test sessions on days 1 and 8 following the same procedure as the conditioning session, but without temperature manipulation. Instead, the same target temperature corresponding to VAS 60 was maintained across all conditions (see *Figure 1* for details of the design). The order of condition was pseudorandomised, and each trial type was repeated ten times in each test session. On day 8, one stimulus per stimulation intensity (i.e. VAS 40, 60, and 80) was applied before the start of the test session to re-familiarise participants with the thermal stimulation. Note that participants were informed that these pre-test stimuli were part of the recalibration and refamiliarisation procedure conducted prior to the second test session.

### Pain intensity ratings

During the conditioning and the test sessions, participants provided pain intensity ratings on a VAS with endpoints points 0 (='not painful at all') and 100 (='unbearably painful'). The cursor was positioned randomly on the scale at the beginning of the rating period. Participants could move the cursor by pressing the left or right arrow key and were asked to confirm their rating with the 'enter' key (no time limit).

### Reaction time task

During the conditioning and test sessions, a reaction time task was included at the beginning of each trial in which participants had to respond as quickly as possible to a target stimulus (a blue cross) by pressing the left arrow key to ensure sustained attention. The blue cross appeared for 300 ms with a jittered onset at the beginning of each trial, that is, 0–5 s after trial onset.

### Psychological questionnaires

Before calibration on day 1, participants completed the German version of the following questionnaires using an online survey system (LimeSurvey, LimeSurvey GmbH, Hamburg, Germany): the Generic Rating for Treatment Pre-Experiences, Treatment Expectations, and Treatment Effects (GEEE; *Rief et al., 2021*), the Somatosensory Amplification Scale (SSAS; *Doering et al., 2015*; *Barsky et al., 1988*), the Perceived Stress Scale (PSS-10; *Cohen et al., 1983*; *Klein et al., 2016*), State-Trait-Anxiety-Depression-Inventory (STADI Trait; *Laux et al., 2013*), and the Pain Catastrophizing Scale (PCS; *Sullivan et al., 1995*; *Meyer et al., 2008*). Warmth and competence of the experimenter were assessed as described in *Seewald and Rief, 2024* at the end of day 1. In short, participants were asked the question how the experimenter seemed to them and provided ratings on a 5-point scale ranging from 1 (=not at all) to 5 (=extremely) for the following descriptors in German: 'friendly', 'well-intentioned', 'trustworthy', 'warm', 'good-natured', and 'sincere' to capture experimenter warmth and 'competent', 'confident', 'capable', 'efficient', 'intelligent', and 'skilful' for experimenter competence. The mean across items of each scale was used in further analyses.

Treatment expectation ratings using the GEEE and the emotional state using STADI State were also collected before conditioning, after conditioning, and before test session 2 on day 8. Treatment effects were rated after conditioning and after test sessions 1 and 2. Note that participants also completed the following questionnaires as part of a larger project: Fear of Pain Questionnaire III, Behavioral Inhibition and Behavioral Activation scales, 10-item Big Five Inventory, and the Positive and Negative Affect Schedule. Responses to these questionnaires will be analysed elsewhere.

### Statistical analyses

Data were analysed using SPSS (version 29.0.2.0). For each of the three conditions, mean pain intensity ratings for the calibration phase, conditioning phase and tests on days 1 and 8 were calculated across the trials of the respective phase. Nocebo effects were defined as the difference in pain intensity ratings between the nocebo and the control condition (nocebo – control), placebo effects as the

difference between the control and the placebo condition (control – placebo). Comparisons of stimulation intensities and pain intensity ratings between conditions were carried out using repeated-measures ANOVAs with the within-subject factor *Condition* (placebo, nocebo, control) followed by post hoc Bonferroni-corrected pairwise comparisons. To compare the magnitude and persistence of placebo and nocebo effects over time, an ANOVA with the within-subject factors *Effect* (placebo effect, nocebo effect) and *Session* (day 1, day 8) was used. The analysis of trial-by-trial ratings used an ANOVA with the within-subject factors *Condition* (placebo, nocebo, control) and *Trial* (trials 1–12). To account for interindividual differences in conditioning, the difference between the pain worsening and pain relief during the conditioning phase was entered as a covariate in the comparison of pain intensity ratings at days 1 and 8 (ANCOVA). Degrees of freedom were corrected using the Greenhouse–Geisser estimate of sphericity. To explore the relationship between placebo and nocebo effects on both test days, we calculated the Pearson correlation coefficient. Because expectancy ratings were not normally distributed, non-parametric Wilcoxon signed rank tests were used to compare these ratings between conditions and timepoints and Spearman's rho was calculated for correlations between pain intensity and expectancy ratings. All questionnaires were analysed according to their respective manuals.

Separate multiple linear regression analyses were performed to examine the influence of expectations (GEEE ratings) and experienced effects (VAS ratings) on subsequent placebo and nocebo effects. For day 1, the placebo effect was entered as the dependent variable and the following variables as potential predictors: (i) expected pain improvement with placebo before conditioning (i.e., placebo expectation), (ii) pain relief during conditioning, and (iii) the expected pain improvement with placebo before the test session at day 1. The equivalent analysis was conducted for the nocebo effect but with (i) expected pain worsening with nocebo before conditioning (i.e., nocebo expectation), (ii) pain worsening during conditioning, and (iii) the expected pain worsening with nocebo before the test session at day 1 as predictors.

To predict placebo responses a week later ($VAS_{control}$ – $VAS_{placebo}$ at day 8), the same independent variables were entered as for day 1 but with the following additional variables: (i) pain ratings during test trials at day 1 and (ii) the expected pain improvement with placebo before the test session at day 8. In the equivalent analysis for the nocebo effect on day 8 as dependent variable, we added (i) the nocebo effect at day 1 and (ii) the expected pain worsening with nocebo before the test session at day 8.

To explore whether psychological variables could explain additional variance in the regression analyses, we repeated all four analyses described above but included scores from these questionnaires as additional independent variables: SSAS, PSS-10, STADI trait, PCS, and experimenter warmth and competence scores.

In all analyses, a significance level of $P<0.05$ was used, and pairwise comparisons were conducted using two-tailed $P$-values. For all multiple regression analyses, the regression coefficient is reported.

## Acknowledgements

The work was funded by the Deutsche Forschungsgemeinschaft (DFG, German Research Foundation), Project-ID 422744262-TRR 289 (gefördert durch die DFG, Projektnummer 422744262, TRR 289).

## Additional information

### Funding

| Funder | Grant reference number | Author |
| --- | --- | --- |
| Deutsche Forschungsgemeinschaft | 422744262 | Ulrike Bingel |

The funders had no role in study design, data collection and interpretation, or the decision to submit the work for publication.

## Author contributions

Angelika Kunkel, Data curation, Formal analysis, Writing – original draft, Project administration, Writing – review and editing; Katharina Schmidt, Conceptualization, Data curation, Formal analysis, Writing – original draft, Project administration, Writing – review and editing; Helena Hartmann, Katja Wiech, Data curation, Formal analysis, Writing – original draft, Writing – review and editing; Torben Strietzel, Investigation; Jens-Lennart Sperzel, Investigation, Writing – original draft; Ulrike Bingel, Conceptualization, Data curation, Supervision, Funding acquisition, Writing – original draft, Project administration, Writing – review and editing

## Author ORCIDs

Angelika Kunkel ⓘ https://orcid.org/0000-0001-8303-8693
Katharina Schmidt ⓘ https://orcid.org/0000-0001-9397-5285
Helena Hartmann ⓘ https://orcid.org/0000-0002-1331-6683
Katja Wiech ⓘ https://orcid.org/0000-0002-5062-1046
Ulrike Bingel ⓘ https://orcid.org/0000-0002-9528-3204

## Ethics

Human subjects: Ethics approval was granted by the University Hospital Essen (22-10597-BO). The experiment adhered to the principles outlined in the 2013 Declaration of Helsinki. Informed written consent was obtained from all participants, who received 120 Euros for their participation.

Reviewer #1 (Public review): https://doi.org/10.7554/eLife.105753.3.sa1
Reviewer #2 (Public review): https://doi.org/10.7554/eLife.105753.3.sa2
Author response https://doi.org/10.7554/eLife.105753.3.sa3

---

## Additional files

### Supplementary files
MDAR checklist

### Data availability
All data generated and analyzed during this study are available in the Open Science Framework.

The following dataset was generated:

| Author(s) | Year | Dataset title | Dataset URL | Database and Identifier |
|---|---|---|---|---|
| Kunkel A, Schmidt K, Hartmann H, Strietzel T, Sperzel JL, Wiech K, Bingel U | 2025 | Nocebo effects are stronger and more persistent than placebo effects in healthy individuals | https://doi.org/10.17605/OSF.IO/QC73U | Open Science Framework, 10.17605/OSF.IO/QC73U |

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

## Appendix 1

### Supplementary material
Supplementary results
Stimulus calibration

The calibration procedure determined two temperature levels which were used in the subsequent conditioning procedure to induce the perception of pain reduction (placebo condition: M = 44.54°C, SD = 1.37) or pain increase (nocebo condition: M = 46.18°C, SD = 1.08). A third level was found for the control condition (M = 45.38°C, SD = 1.20). The three temperature levels were perceived as significantly different (VAS ratings; placebo condition: M = 32.90, SD = 16.17; nocebo condition: M = 80.84, SD = 12.18; control condition: M = 56.62, SD = 17.09; F(1.95, 200.86) = 466.70, P<0.001; control vs. placebo condition: t(103) = 15.85, P<0.001, 95% CI, 20.74–26.68; nocebo vs. placebo condition: t(103) = 28.37, P<0.001, 95% CI, 44.58–51.28; nocebo vs. control condition: t(103) = 16.0, P<0.001, 95% CI, 21.22–27.22). Importantly, pain intensity ratings for placebo and the nocebo stimuli were both equidistant from the control condition (control minus placebo: M = 23.71, SD = 15.26; nocebo minus control: M = 24.22, SD = 15.44; 95% CI, –5.45–4.43; t(103) = –0.21, P = 0.84), indicating successful stimulus calibration in both directions.

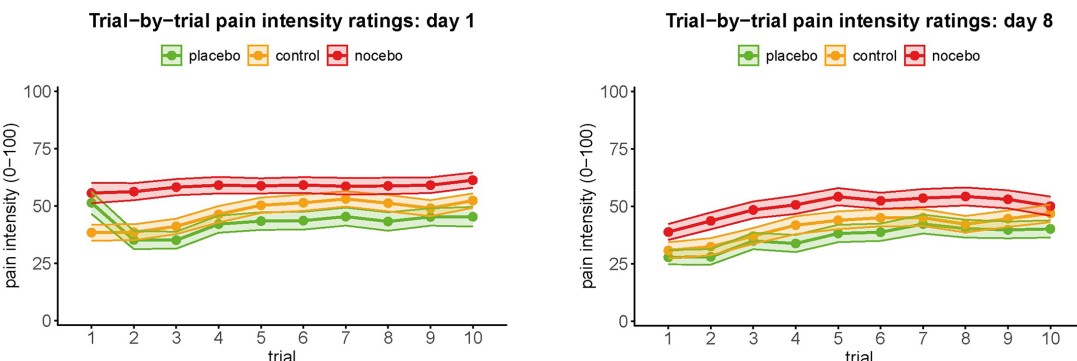

**Appendix 1—figure 1.** Trial-by-trial pain intensity rating during the test phases of days 1 and 8.

**Appendix 1—table 1.** Results of multiple linear regression analyses with expectations and prior experience as independent variables and placebo effect or nocebo effect as the dependent variable.

| | Beta | Std error | t-value | P-value |
|---|---|---|---|---|
| **Day 1: placebo effect** | | | | |
| *Intercept* | –2.919 | 3.756 | –0.777 | 0.439 |
| Expected improvement pre-conditioning (GEEE) | –0.517 | 0.736 | –0.702 | 0.485 |
| Conditioning placebo effect (VAS rating) | 0.300 | 0.094 | 3.179 | **0.002**** |
| Expected improvement on day 1 (GEEE) | 0.479 | 0.571 | 0.839 | 0.404 |
| **Day 1: nocebo effect** | | | | |
| *Intercept* | 3.261 | 3.539 | 0.921 | 0.359 |
| Expected worsening pre-conditioning (GEEE) | 0.288 | 0.446 | 0.646 | 0.519 |
| Conditioning nocebo effect (VAS rating) | 0.136 | 0.075 | 1.807 | 0.074 |
| Expected worsening on day 1 (GEEE) | 0.355 | 0.425 | 0.837 | 0.405 |
| **Day 8: placebo effect** | | | | |
| *Intercept* | 3.531 | 2.478 | 1.425 | 0.157 |
| Expected improvement pre-conditioning (GEEE) | –0.589 | 0.474 | –1.243 | 0.217 |
| Conditioning placebo effect (VAS rating) | 0.083 | 0.066 | 1.244 | 0.217 |

*Appendix 1—table 1 Continued on next page*

*Appendix 1—table 1 Continued*

| | Beta | Std error | *t*-value | *P*-value |
|---|---|---|---|---|
| Expected improvement on day 1 (GEEE) | –0.769 | 0.463 | –1.661 | 0.100 |
| Placebo effect on day 1 (VAS rating) | 0.315 | 0.064 | 4.904 | <0.001*** |
| Expected improvement on day 8 (GEEE) | 1.033 | 0.459 | 2.251 | 0.027* |
| **Day 8: nocebo effect** | | | | |
| *Intercept* | 1.253 | 3.826 | 0.328 | 0.744 |
| Expected worsening pre-conditioning (GEEE) | 0.187 | 0.513 | 0.364 | 0.716 |
| Conditioning nocebo effect (VAS rating) | 0.144 | 0.083 | 1.732 | 0.087 |
| Expected worsening on day 1 (GEEE) | –0.395 | 0.523 | –0.755 | 0.452 |
| Nocebo effect on day 1 (VAS rating) | 0.272 | 0.110 | 2.482 | 0.015* |
| Expected worsening on day 8 (GEEE) | 0.210 | 0.457 | 0.459 | 0.647 |

GEEE, Generic Rating Scale for Previous Treatment Experiences, Treatment Expectations, and Treatment Effect; VAS, visual analogue scale. ***Rief et al., 2021***.
***P<0.001, **P<0.01, *P<0.05.

**Appendix 1—table 2.** Results of multiple linear regression analyses with expectations, prior experience and psychological factors as independent variables and placebo effect or nocebo effect as the dependent variable.

| | Beta | Std error | *t*-value | *P*-value |
|---|---|---|---|---|
| **Day 1: placebo effect** | | | | |
| *Intercept* | 1.294 | 22.168 | | |
| Expected improvement pre-conditioning (GEEE) | –0.531 | 0.750 | –0.708 | 0.481 |
| Conditioning placebo effect (VAS rating) | 0.303 | 0.097 | 3.120 | 0.002** |
| Expected improvement on day 1 (GEEE) | 0.626 | 0.575 | 1.088 | 0.280 |
| **Psychological variables:** | | | | |
| Somatosensory amplification | –0.654 | 0.227 | –2.877 | 0.005** |
| Trait anxiety | 0.042 | 0.334 | 0.126 | 0.900 |
| Trait depression | 0.220 | 0.308 | 0.712 | 0.478 |
| Behavioural inhibition | –1.528 | 3.100 | –0.493 | 0.623 |
| Behavioural activation | 4.922 | 3.299 | 1.492 | 0.139 |
| Pain catastrophising | 0.070 | 0.136 | 0.511 | 0.611 |
| Experimenter warmth | –4.161 | 3.027 | –1.375 | 0.173 |
| Experimenter competence | 4.732 | 3.742 | 1.265 | 0.209 |
| **Day 1: nocebo effect** | | | | |
| *Intercept* | –36.384 | 15.752 | –2.310 | 0.023 |
| Expected worsening pre-conditioning (GEEE) | 0.402 | 0.470 | 0.856 | 0.394 |
| Conditioning nocebo effect (VAS rating) | 0.131 | 0.078 | 1.690 | 0.095 |
| Expected worsening on day 1 (GEEE) | 0.200 | 0.442 | 0.454 | 0.651 |
| **Psychological variables:** | | | | |
| Somatosensory amplification | –0.206 | 0.167 | –1.235 | 0.220 |
| Trait anxiety | 0.445 | 0.255 | 1.745 | 0.084 |
| Trait depression | –0.096 | 0.225 | –0.425 | 0.672 |
| Behavioural inhibition | 4.011 | 2.312 | 1.735 | 0.086 |

*Appendix 1—table 2 Continued on next page*

*Appendix 1—table 2 Continued*

| | Beta | Std error | *t*-value | *P*-value |
|---|---|---|---|---|
| Behavioural activation | 4.800 | 2.416 | 1.987 | 0.050 |
| Pain catastrophising | 0.084 | 0.098 | 0.857 | 0.394 |
| Experimenter warmth | –0.352 | 2.314 | –0.152 | 0.879 |
| Experimenter competence | 5.791 | 2.801 | 2.067 | **0.042*** |
| **Day 8: placebo effect** | | | | |
| *Intercept* | 17.843 | 14.498 | 1.231 | 0.222 |
| Expected improvement pre-conditioning (GEEE) | –0.702 | 0.496 | –1.417 | 0.160 |
| Conditioning placebo effect (VAS rating) | 0.063 | 0.073 | 0.863 | 0.391 |
| Expected improvement on day 1 (GEEE) | –0.949 | 0.507 | –1.872 | 0.065 |
| Placebo effect on day 1 (VAS rating) | 0.382 | 0.071 | 5.343 | **<0.001*** |
| Expected improvement on day 8 (GEEE) | 1.131 | 0.536 | 2.110 | **0.038*** |
| Psychological variables: | | | | |
| Somatosensory amplification | 0.233 | 0.167 | 1.396 | 0.167 |
| Trait anxiety | –0.215 | 0.227 | –0.948 | 0.346 |
| Trait depression | –0.217 | 0.218 | –0.995 | 0.323 |
| Behavioural inhibition | –1.344 | 2.051 | –0.655 | 0.514 |
| Behavioural activation | –1.166 | 2.232 | –0.522 | 0.603 |
| Pain catastrophising | –0.129 | 0.092 | –1.403 | 0.165 |
| Experimenter warmth | –0.727 | 2.059 | –0.353 | 0.725 |
| Experimenter competence | –0.563 | 2.499 | –0.225 | 0.822 |
| **Day 8: nocebo effect** | | | | |
| *Intercept* | –9.215 | 17.887 | –0.515 | 0.608 |
| Expected worsening pre-conditioning (GEEE) | 0.650 | 0.580 | 1.122 | 0.265 |
| Conditioning nocebo effect (VAS rating) | 0.103 | 0.092 | 1.119 | 0.267 |
| Expected worsening on day 1 (GEEE) | –0.325 | 0.565 | –0.574 | 0.567 |
| Nocebo effect on day 1 (VAS rating) | 0.290 | 0.124 | 2.349 | **0.021*** |
| Expected worsening on day 8 (GEEE) | –0.154 | 0.516 | –0.298 | 0.766 |
| Psychological variables: | | | | |
| Somatosensory amplification | –0.080 | 0.199 | –0.403 | 0.688 |
| Trait anxiety | 0.099 | 0.288 | 0.345 | 0.731 |
| Trait depression | 0.093 | 0.261 | 0.356 | 0.723 |
| Behavioural inhibition | 1.759 | 2.600 | 0.677 | 0.501 |
| Behavioural activation | –1.615 | 2.861 | –0.565 | 0.574 |
| Pain catastrophising | 0.064 | 0.110 | 0.576 | 0.566 |
| Experimenter warmth | 2.569 | 2.687 | 0.956 | 0.342 |
| Experimenter competence | –0.608 | 3.190 | –0.191 | 0.849 |

GEEE, Generic Rating Scale for Previous Treatment Experiences, Treatment Expectations, and Treatment Effect, VAS, visual analogue scale. ***Rief et al., 2021***
***P<0.001, **P<0.01, *P<0.05.

