## [Editor Report · eLife Assessment]

In this preregistered study, Kunkel and colleagues set out to compare the magnitude and duration of placebo versus nocebo effects in healthy volunteers, and also to examine the different factors contributing to these effects. The authors follow a rigorous methodology in a within-subjects design, taking into consideration standard conventions for manipulation of expectations, and using an appropriate sham condition. They present **compelling** evidence of long-lasting placebo and nocebo effects, with nocebo responses demonstrating consistently greater strength. These **valuable** results have the potential for a great impact in the field of experimental and clinical pain.

---

## [Referee Report · Reviewer #1 (Public review)]

Summary:

The study aimed to: (1) assess the magnitude of placebo and nocebo effects immediately after induction through verbal instructions and conditioning, (2) examine the persistence of these effects one week later, and (3) identify predictors of sustained placebo and nocebo responses over time.

Strengths:

An innovation was to use sham TENS stimulation as the expectation manipulation. This expectation manipulation was reinforced not only by the change in pain stimulus intensity, but also by delivery of non-painful electrical stimulation, labelled as TENS stimulation.

Questionnaire-based treatment expectation ratings were collected before conditioning and after conditioning, and after the test session, which provided an explicit measure of participant's expectations about the manipulation.

The finding that placebo and nocebo effects are influenced by recent experience provides a novel insight into a potential moderator of individual placebo effects.

Weaknesses:

There are a limited number of trials per test condition (10) which means that the trajectory of responses to the manipulation may not be explored, which would be an interesting future study.

The differences between the nocebo and control condition in pain ratings during conditioning could be explained by differing physiological effects of the different stimulus intensities, so it is difficult to make any claims about the expectation effects here.A a randomisation error meant that 25 participants received an unbalanced number 448 of trials per condition (i.e., 10 x VAS 40, 14 x VAS 60, 12 x VAS 80), although the authors accounted for this during analysis so it is not of major concern.

This manuscript presents a study on expectation manipulation to induce placebo and nocebo effects in healthy participants. The study follows standard placebo experiment conventions with use of TENS stimulation as the placebo manipulation. The authors were able to achieve their aims. A key finding is that placebo and nocebo effects were predicted by recent experience, which is a novel contribution to the literature. The findings provide insights into the differences between placebo and nocebo effects and the potential moderators of these effects.

Comments on revisions:

I am satisfied with the author's revisions to the manuscript and have no further comments.

---

## [Referee Report · Reviewer #2 (Public review)]

Summary:

Kunkel et al aim to answer a fundamental question: Do placebo and nocebo effects differ in magnitude or longevity? To address this question, they used a powerful within-participants design, with a very large sample size (n=104), in which they compared placebo and nocebo effects - within the same individuals - across verbal expectations, conditioning, testing phase, and a 1-week follow-up. With elegant analyses, they establish that different mechanisms underlie the learning of placebo vs nocebo effects, with the latter being acquired faster and extinguished slower. This is an important finding for both the basic understanding of learning mechanisms in humans and for potential clinical applications to improve human health.

Strengths:

Beyond the above - the paper is well-written and very clear. It lays out nicely the need for the current investigation and what implications it holds. The design is elegant, and the analyses are rich, thoughtful, and interesting. The sample size is large which is highly appreciated, considering the longitudinal, in-lab study design. The question is super important and well-investigated, and the entire manuscript is very thoughtful with analyses closely examining the underlying mechanisms of placebo versus nocebo effects.

Comments on revisions:

The authors have addressed all of my concerns and comments - one point for them to verify is that indeed analyses that have not been preregistered will be flagged as such. The provided pre-registration link doesn't specify much about the analysis plans and specific tests used.

---

## [Author Response]

The following is the authors’ response to the original reviews

**Public Reviews:**

**Reviewer #1 (Public review):**
Summary:This manuscript presents a study on expectation manipulation to induce placebo and nocebo effects in healthy participants. The study follows standard placebo experiment conventions with the use of TENS stimulation as the placebo manipulation. The authors were able to achieve their aims. A key finding is that placebo and nocebo effects were predicted by recent experience, which is a novel contribution to the literature. The findings provide insights into the differences between placebo and nocebo effects and the potential moderators of these effects.Specifically, the study aimed to:(1) assess the magnitude of placebo and nocebo effects immediately after induction through verbal instructions and conditioning(2) examine the persistence of these effects one week later, and(3) identify predictors of sustained placebo and nocebo responses over time.Strengths:An innovation was to use sham TENS stimulation as the expectation manipulation. This expectation manipulation was reinforced not only by the change in pain stimulus intensity, but also by delivery of non-painful electrical stimulation, labelled as TENS stimulation.Questionnaire-based treatment expectation ratings were collected before conditioning and after conditioning, and after the test session, which provided an explicit measure of participants' expectations about the manipulation.The finding that placebo and nocebo effects are influenced by recent experience provides a novel insight into a potential moderator of individual placebo effects.

We thank the reviewer for their thorough evaluation of our manuscript and for highlighting the novelty and originality of our study.

Weaknesses:There are a limited number of trials per test condition (10), which means that the trajectory of responses to the manipulation may not be adequately explored.

We appreciate the reviewer’s comment regarding the number of trials in the test phase. The trial number was chosen to ensure comparability with previous studies addressing similar research questions with similar designs (e.g. Colloca et al., 2010). Our primary objective was to directly compare placebo and nocebo effects within a within-subject design and to examine their persistence one week after the first test session. While we did not specifically aim to investigate the trajectory of responses within a single testing session, we fully agree that a comprehensive analysis of the trajectories of expectation effects on pain would be a valuable extension of our work. We have now acknowledged this limitation and future direction in the revised manuscript.

The paragraph reads as follows:“It is important to note that our study was designed in alignment with previous studies addressing similar questions (e.g., Colloca et al., 2010). Our primary aim was to directly compare placebo and nocebo effects in a within-subject design and assess their persistence of these effects one week following the first test session. One limitation of our approach is the relatively short duration of each session, which may have limited our ability to examine the trajectory of responses within a single session. Future studies could address this limitation by increasing the number of trials for a more comprehensive analysis.”

On day 8, one stimulus per stimulation intensity (i.e., VAS 40, 60, and 80) was applied before the start of the test session to re-familiarise participants with the thermal stimulation. There is a potential risk of revealing the manipulation to participants during the re-familiarization process, as they were not previously briefed to expect the painful stimulus intensity to vary without the application of sham TENS stimulation.

We thank the reviewer for the opportunity to clarify this point. Participants were informed at the beginning of the experiment that we would use different stimulation intensities to re-familiarize them with the stimuli before the second test session. We are therefore confident that participants perceived this step as part of a recalibration rather than associating it with the experimental manipulation. We have added this information to the revised version of the manuscript.

The paragraph now reads as follows:“On day 8, one stimulus per stimulation intensity (i.e., VAS 40, 60 and 80) was applied before the start of the test session to re-familiarise participants with the thermal stimulation. Note that participants were informed that these pre-test stimuli were part of the recalibration and refamiliarization procedure conducted prior to the second test session.”

The differences between the nocebo and control conditions in pain ratings during conditioning could be explained by the differing physiological effects of the different stimulus intensities, so it is difficult to make any claims about expectation effects here.

We appreciate the reviewer’s comment and agree that, despite the careful calibration of the three pain stimuli, we cannot entirely rule out the possibility that temporal dynamics during the conditioning session were influenced by differential physiological effects of the varying stimulus intensities (e.g., intensity-dependent habituation or sensitization). We have addressed this in the revision of the manuscript, but we would like to emphasize that the stronger nocebo effects during the test phase are statistically controlled for any differences in the conditioning session.

The paragraph now reads:“This asymmetry is noteworthy in and of itself because it occurred despite the equidistant stimulus calibration relative to the control condition prior to conditioning. It may be the result of different physiological effects of the stimuli over time or amplified learning in the nocebo condition, consistent with its heightened biological relevance, but it could also be a stronger effect of the verbal instructions in this condition.”

A randomisation error meant that 25 participants received an unbalanced number of 448 trials per condition (i.e., 10 x VAS 40, 14 x VAS 60, 12 x VAS 80).

We agree that this is indeed unfortunate. However, we would like to point out that all analyses reported in the manuscript have been controlled for the VAS ratings in the conditioning session, i.e., potential effects of the conditioned placebo and nocebo stimuli. Moreover, we have now conducted additional analyses, presented here in our response to the reviewers, to demonstrate that this imbalance did not systematically bias the results. Importantly, the key findings observed during the test phase remain robust despite this issue.

Specifically, when excluding these 25 participants from the analyses, the reported stronger nocebo compared to placebo effects in the test session on day 1 remain unchanged. Likewise, the comparison of placebo and nocebo effects between days 1 and 8 shows the same pattern when excluding the participants in question. The only exception is the interaction between effect (placebo vs nocebo) x session (day 1 vs day 8), which changed from a borderline significant result (p = .049) to insignificant (p = .24). However, post hoc tests continued to show the same pattern as originally reported: a significant reduction in the nocebo effect from day 1 to day 8 and no significant change in the placebo effect.

**Reviewer #2 (Public review):**
Summary:Kunkel et al aim to answer a fundamental question: Do placebo and nocebo effects differ in magnitude or longevity? To address this question, they used a powerful within-participants design, with a very large sample size (n=104), in which they compared placebo and nocebo effects - within the same individuals - across verbal expectations, conditioning, testing phase, and a 1-week follow-up. With elegant analyses, they establish that different mechanisms underlie the learning of placebo vs nocebo effects, with the latter being acquired faster and extinguished slower. This is an important finding for both the basic understanding of learning mechanisms in humans and for potential clinical applications to improve human health.Strengths:Beyond the above - the paper is well-written and very clear. It lays out nicely the need for the current investigation and what implications it holds. The design is elegant, and the analyses are rich, thoughtful, and interesting. The sample size is large which is highly appreciated, considering the longitudinal, in-lab study design. The question is super important and well-investigated, and the entire manuscript is very thoughtful with analyses closely examining the underlying mechanisms of placebo versus nocebo effects.

We thank the reviewer for their positive evaluation of our manuscript and for acknowledging the methodological rigor and the significant implications for clinical applications and the broader research field.

Weaknesses:There were two highly addressable weaknesses in my opinion:(1) I could not find the preregistration - this is crucial to verify what analyses the authors have committed to prior to writing the manuscript. Please provide a link leading directly to the preregistration - searching for the specified number in the suggested website yielded no results.

We thank the reviewer for pointing this out. We included a link to the preregistration in the revised manuscript. This study was pre-registered with the German Clinical Trial Register (registration number: DRKS00029228; https://drks.de/search/de/trial/DRKS00029228).

(2) There is a recurring issue which is easy to address: because the Methods are located after the Results, many of the constructs used, analyses conducted, and even the main placebo and nocebo inductions are unclear, making it hard to appreciate the results in full. I recommend finding a way to detail at the beginning of the results section how placebo and nocebo effects have been induced. While my background means I am familiar with these methods, other readers will lack that knowledge. Even a short paragraph or a figure (like Figure 4) could help clarify the results substantially. For example, a significant portion of the results is devoted to the conditioning part of the experiment, while it is unknown which part was involved (e.g., were temperatures lowered/increased in all trials or only in the beginning).

We thank the reviewer for their helpful comment and agree that the Results section requires additional information that would typically be provided by the Methods section if it directly followed the Introduction. In response, we have moved the former Figure 4 from the Methods section to the beginning of the Results section as a new Figure 1, to improve clarity. Further, we have revised the Methods section to explicitly state that all trials during the conditioning phase were manipulated in the same way.

**Recommendations for the Authors:**

**Reviewer #1 (Recommendations for the authors):**
(1) Given that the authors are claiming (correctly) that there is only limited work comparing placebo/nocebo effects, there are some papers missing from their citations:Nocebo responses are stronger than placebo responses after subliminal pain conditioning - - Jensen, K., Kirsch, I., Odmalm, S., Kaptchuk, T. J. & Ingvar, M. Classical conditioning of analgesic and hyperalgesic pain responses without conscious awareness. Proc. Natl. Acad. Sci. USA 112, 7863-7 (2015)

We thank the reviewer and have now included this relevant publication into the introduction of the revised manuscript.

Hird, E.J., Charalambous, C., El-Deredy, W. et al. Boundary effects of expectation in human pain perception. Sci Rep 9, 9443 (2019). https://doi.org/10.1038/s41598-019-45811-x

We thank the reviewer for suggesting this relevant publication. We have now included it into the discussion of the revised manuscript by adding the following paragraph:

“Recent work using a predictive coding framework further suggests that nocebo effects may be less susceptible to prediction error than placebo effects (Hird et al., 2019), which could contribute to their greater persistence and strength in our study.”

(2) The trial-by-trial pain ratings could have been usefully modelled with a computational model, such as a Bayesian model (this is especially pertinent given the reference to Bayesian processing in the discussion). A multilevel model could also be used to increase the power of the analysis. This is a tentative suggestion, as I appreciate it would require a significant investment of time and work - alternatively, the authors could acknowledge it in the Discussion as a useful future avenue for investigation, if this is preferred.

We thank the reviewer for this thoughtful suggestion. While we agree that computational modelling approaches could provide valuable insights into individual learning, our study was not designed with this in mind and the relatively small number of trials per condition and the absence of trial-by-trial expectancy ratings limit the applicability of such models. We have therefore chosen not to pursue such analysis but highlight it in the discussion as a promising direction for future research.

“Notably, the most recent experience was the most predictive in all three analyses; for instance, the placebo effect on day 8 was predicted by the placebo effect on day 1, not by the initial conditioning. This finding supports the Bayesian inference framework, where recent experiences are weighted more heavily in the process of model updating because they are more likely to reflect the current state of the environment, providing the most relevant and immediate information needed to guide future actions and predictions24. Interestingly, while a change in pain predicted subsequent nocebo effects, it seemed less influential than for placebo effects. This aligns with findings that longer conditioning enhanced placebo effects, while it did not affect nocebo responses10 and the conclusion that nocebo instruction may be sufficient to trigger nocebo responses. Using Bayesian modeling, future studies could identify individual differences in the development of placebo and nocebo effects by integrating prior experiences and sensory inputs, providing a probabilistic framework for understanding the underlying mechanisms.”

(3) The paper is missing any justification of sample size, i.e. power analysis - please include this.

We apologize for the missing information on our a priori power analysis. As there is a lack of prior studies investigating within-subjects comparisons of placebo and nocebo effects that could inform precise effect size estimates for our research question, we based our calculation on the ability detect small effects. Specifically, the study was powered to detect effect sizes in the range of d = 0.2 - 0.25 with α = .05 and power = .9, yielding a required sample size of N = 83-129. We have now added this information to the methods section of the revised manuscript.

(4) "On day 8, one stimulus per stimulation intensity (i.e., VAS 40, 60 and 80) was applied before the start of the test session to re-familiarise participants with the thermal stimulation."What were the instructions about this? Was it before the electrode was applied? This runs the risk of unblinding participants, as they only expect to feel changes in stimulus intensity due to the TENS stimulation.

We thank the reviewer for pointing out the potential risk of unblinding participants due to the re-familiarization process prior to the second test session. We would like to clarify that we followed specific procedures to prevent participants from associating this process with the experimental manipulation. The re-familiarisation with the thermal stimuli was conducted after the electrode had been applied and re-tested to ensure that both stimulus modalities were re-introduced in a consistent and neutral context. Participants were explicitly informed that both procedures were standard checks prior to the actual test session (“We will check both once again before we begin the actual measurement.”). For the thermal stimuli, we informed participants that they would experience three different intensities to allow the skin to acclimate (e.g., “...we will test the heat stimuli in 3 trials with different temperatures, allowing your skin to acclimate to the stimuli. …”), without implying any connection to the experimental conditions.

Importantly, this re-familiarization procedure mirrored what participants had already experienced during the initial calibration session on day 1. We therefore assume that participants interpreted as a routine technical step rather than part of the experimental manipulation. We have now clarified this procedure in the methods section of the revised manuscript.

(5) "For a comparison of pain intensity ratings between time-points, an ANOVA with the within-subject factors Condition (placebo, nocebo, control) and Session (day 1, day 8) was carried out. For the comparison of placebo and nocebo effects between the two test days, an ANOVA with the with-subject factors Effect (placebo effect, nocebo effect) and Session (day 1, day 8) was used."It seems that one ANOVA is looking at raw pain scores and one is looking at difference scores, but this is a bit confusing - please rephrase/clarify this, and explain why it is useful to include both.

We thank the reviewer for highlighting this point. Our primary analyses focus on placebo and nocebo effects, which we define as the difference in pain intensity ratings between the control and the placebo condition (placebo effect) and the nocebo and the control condition (nocebo effect), respectively.

To examine whether condition effects were present at each time-point, we first conducted two separate repeated measures ANOVAs - one for day 1 and one for day 8 - with the within-subject factor CONDITION (placebo, nocebo, control).

To compare the magnitude and persistence of placebo and nocebo effects over time, we then calculated the above-mentioned difference scores and submitted these to a second ANOVA with within-subject factors EFFECT (placebo vs. nocebo effect) and SESSION (day 1 vs. day 8). We have now clarified this approach on page 19 of the revised manuscript. To avoid confusion, the Condition x Session ANOVA has been removed from the manuscript.

(6) Please can the authors provide a figure illustrating trial-by-trial ratings during test trials as well as during conditioning trials?

In response to the reviewer’s point, we now provide the trial-by-trial ratings of the test phases on days 1 and 8 as an additional figure in the Supplement (Figure S1) and would like to clarify that trial-by-trial pain intensity ratings of the conditioning phase are displayed in Figure 2C of the manuscript,

(7) "Separate multiple linear regression analyses were performed to examine the influence of expectations (GEEE ratings) and experienced effects (VAS ratings) on subsequent placebo and nocebo effects. For day 1, the placebo effect was entered as the dependent variable and the following variables as potential predictors: (i) expected improvement with placebo before conditioning, (ii) placebo effect during conditioning and (iii) the expected improvement with placebo before the test session at day 1"The term "placebo effect during conditioning" is a bit confusing - I believe this is just the effect of varying stimulus intensities - please could the authors be more explicit on the terminology they use to describe this? NB changes in pain rating during the conditioning trials do not count as a placebo/nocebo effect, as most of the change in rating will reflect differences in stimulation intensity.

We agree with the reviewer that the cited paragraph refers to the actual application of lower or higher pain stimuli during the conditioning session, rather than genuinely induced placebo or nocebo effect. We thank the reviewer for this helpful observation and have revised the terminology, accordingly, now referring to these as “pain relief during conditioning” and “pain worsening during conditioning”.

(8) Supplementary materials: "The three temperature levels were perceived as significantly different (VAS ratings; placebo condition: M = 32.90, SD = 16.17); nocebo condition: M = 56.62, SD = 17.09; control condition: M = 80.84, SD = 12.18"This suggests that the VAS rating for the control condition was higher than for the nocebo condition. Please could the authors clarify/correct this?

We thank the reviewer for spotting this error. The values for the control and the nocebo condition had accidentally been swapped. This has now been corrected in the manuscript: *control condition: M = 56.62, SD = 17.09; nocebo condition: M = 80.84, SD = 12.18*.

(9) "To predict placebo responses a week later (VAScontrol - VASplacebo at day 8), the same independent variables were entered as for day 1 but with the following additional variables (i) the placebo effect at day 1 and (ii) the expected improvement with placebo before the test session at day 8."Here it would be much clearer to say 'pain ratings during test trials at day 1".

We agree with the reviewer and have revised the manuscript as suggested.

(10) For completeness, please present the pain intensity ratings during conditioning as well as calibration/test trials in the figure.

Please see our answer to comment (6).

(11) In Figure 1a, it looks like some participants had rated the control condition as zero by day 8. If so, it's inappropriate to include these participants in the analysis if they are not responding to the stimulus. Were these the participants who were excluded due to pain insensitivity?

On day 8, the lowest pain intensity ratings observed were VAS 3 in the placebo condition and VAS 2 in the control condition, both from the same participant. All other participants reported minimum values of VAS 11 or higher (all on a scale from 0-100). Thus, no participant provided a pain rating of VAS 0, and all ratings indicated some level of pain perception in response to the stimulus. We did not define an exclusion criterion based on day 8 pain ratings in our preregistration, and we did not observe any technical issues with the stimulation procedure. To avoid post-hoc exclusions and maintain consistency with our preregistered analysis plan, we therefore decided to include all participants in the analysis.

(12) "Comparison of day 1 and day 8. A direct comparison of placebo and nocebo effects on day 1 and day 8 pain intensity ratings showed a main effect of Effect with a stronger nocebo effect (F(1,97) = 53.93, 131 p< .001, η2 = .36) but no main effect of Day (F(1,97) = 2.94, p = .089, η2 = .029). The significant Effect x Session interaction indicated that the placebo effect and the nocebo effect developed differently over time (F(1,97) = 3.98, p = .049, η2 = .039)"This is confusing as it talks about a main effect of "day" and then interaction with "session" - are they two different models? The authors need to clarify.

We thank the reviewer for pointing this out. In our analysis, “Session” is the correct term for the experimental factor, which has two factor levels, “day 1” and “day 8”. This has now been corrected in the revised manuscript.

**Reviewer #2 (Recommendations for the authors):**
(1) More information on how "size of the effect" in Figures 1b and 2b was calculated is needed; this can be in the legend. If these are differences between control and each condition, then they were reversed for one condition (nocebo?), which is ok - but this should be clearly explained.

We agree with the reviewer and have now revised the figure legends to improve clarity. The legends now read:

1b: “Figure 1. Pain intensity ratings and placebo and nocebo effects during calibration and test sessions. (A) Mean pain intensity ratings in the placebo, nocebo and control condition during calibration, and during the test sessions at day 1 and day 8. (B) Placebo effect (control condition - placebo condition, i.e., positive value of difference) and nocebo effect (nocebo condition - control condition, i.e., positive value of difference) on day 1 and day 8. Error bars indicate the standard error of the mean, circles indicate mean ratings of individual participants. ***: *p* < .001, **: *p* < .01, n.s.: non-significant.”

2b: “Figure 2. Mean and trial-by-trial pain intensity ratings, placebo and nocebo effects during conditioning. (A) Mean pain intensity ratings of the placebo, nocebo and control condition during conditioning. (B) Placebo effect (control condition - placebo condition, i.e., positive value of difference) and nocebo effect (nocebo condition - control condition, i.e., positive value of difference) during conditioning. (C) Trial-by-trial pain intensity ratings (with confidence intervals) during conditioning. Error bars indicate the standard error of the mean, circles indicate mean ratings of individual participants. ***: *p* < .001.”

(2) In the methods, I was missing a clear understanding of how many trials there were in the conditioning phase, and then how many in the other testing phases. Also, how long did the experiment last in total?

We apologize that the exact number of trials in the testing phases was not clear in the original manuscript. We now indicate on page 18 of the revised manuscript that we used 10 trials per condition in the test sessions. We have also added information on the duration of each test day (i.e., three hours on day 1 and one hour on day 8) on page 15.

(3) In expectancy ratings, line 186 - are improvement and worsening expectations different from expected pain relief? It is implied that these are two different constructs - it would be helpful to clarify that.

We agree that this is indeed confusing and would like to clarify that both refer to the same construct. We used the Generic rating scale for previous treatment experiences, treatment expectations, and treatment effects (GEEE questionnaire, Rief et al. 2021) that discriminates between expected symptom improvement, expected symptom worsening, and expected side effects due to a treatment. We now use the terms “expected pain relief” and “expected pain worsening” throughout the whole manuscript.

(4) In the last section of the Results, somatosensory amplification comes out of nowhere - and could be better introduced (see point 2 above).

We agree with the reviewer that introducing the concept of somatosensory amplification and its potential link to placebo/nocebo effects only in the Methods is unhelpful, given that this section appears at the end of the manuscript. We therefore now introduce the relevant publication (Doering et al., 2015) before reporting our findings on this concept.

(5) In line 169, if the authors want to specify what portion of the variance was explained by expectancy, they could conduct a hierarchical regression, where they first look at R2 without the expectancy entered, and only then enter it to obtain the R2 change.

We fully agree that hierarchical regression can be a useful approach for isolating the contribution of variables. However, in our case, expectancy was assessed at different time points (e.g., before conditioning and before the test session on day 1), and there was no principled rationale for determining the order in which these different expectancy-related variables should be entered into a hierarchical model.

That said, in response to the reviewer’s suggestion, we have now conducted hierarchical regression analyses in which all expectancy-related variables were entered together as a single block (see below). These analyses largely confirmed the findings reported so far and are provided here in the response to the reviewers below. Given the exploratory nature of this grouping and the lack of an a priori hierarchy, we feel that the standard multiple regression models remain the most appropriate for addressing our research question because it allows us to evaluate the total contribution of expectancy-related predictors while also examining the individual contribution of each variable within the block. We would therefore prefer to retain these as the primary analyses in the manuscript.

Results of the hierarchical regression analyses:

Day 1 - Placebo response: In step 1, we entered the difference in pain intensity ratings between the control and the placebo condition during conditioning as a predictor. In step 2, we added the two variables reflecting expectations (i.e., expected improvement with placebo (i) before conditioning and (ii) before the test session on day 1). This allowed us to assess whether expectation-related variables explained additional variance beyond the effect of conditioning.

The overall regression model at step 1 was significant, F(1, 102) = 13.42, p < .001, explaining 11.6% of the variance in the dependent variable (R^2^ = .116). Adding the expectancy-related predictors in step 2 did not lead to a significant increase in explained variance, ΔR^2^ = .007, F(2, 100) = 0.384, p = .682. Thus, the conditioning response significantly predicted placebo-related pain reduction on day 1, but additional information on expectations did not account for further variance.

Day 1 - Nocebo response: The equivalent analysis was run for the nocebo response on day 1. In step 1, the pain intensity difference between the nocebo and the control condition was entered as a predictor before adding the two expectancy ratings (i.e., expected worsening with nocebo (i) before conditioning and (ii) before the test session on day 1).

In step 1, the regression model was not statistically significant, F(1, 102) = 2.63, p = .108, and explained only 2.5% of the variance in nocebo response (R^2^ = .025). Adding the expectation-related predictors in Step 2 slightly increased the explained variance by ΔR^2^ = .027, but this change was also non-significant, F(2, 100) = 1.41, p = .250. The overall variance explained by the full model remained low (R^2^ = .052). These results suggest that neither conditioning nor expectation-related variables reliably predicted nocebo-related pain increases on day 1.

Day 8 - Placebo response: For the prediction of the placebo effect on day 8, the following variables reflecting perceived effects were entered as predictors in step 1: the difference in pain intensity ratings between the control and the placebo condition (i) during conditioning and (ii) on day 1. In step 2, the variables reflecting expectations were added: the expected improvement with placebo (i) before conditioning, (ii) before the test session on day 1 and (iii) before the test session on day 8.

In step 1, the model was statistically significant, F(3, 95) = 14.86, p < .001, explaining 23.8% of the variance in the placebo response (R^2^ = .238, Adjusted R^2^ = .222). In step 2, the addition of the expectation-related predictors resulted in a non-significant improvement in model fit, ΔR^2^ = .051, F(3, 92) = 2.21, p = .092. The overall variance explained by the full model increased modestly to 29.0%.

Day 8 - Nocebo response: For the equivalent analyses of nocebo responses on day 8, the following variables were included in step 1: the difference in pain intensity ratings between the nocebo and the control condition (i) during conditioning and (ii) on day 1. In step 2, we entered the variables reflecting nocebo expectations including expected worsening with nocebo (i) before conditioning, (ii) before the test session on day 1 and (iii) before the test session on day 8.In step 1, the model significantly predicted the day 8 nocebo response, F(3, 95) = 6.04, p = .003, accounting for 11.3% of the variance (R^2^ = .113, Adjusted R^2^ = .094). However, the addition of expectation-related predictors in Step 2 resulted in only a negligible and non-significant improvement, ΔR^2^ = .006, F(3, 92) = 0.215, p = .886. The full model explained just 11.9% of the variance (R^2^ = .119).

Typos:(6) Abstract - 104 heathy xxx (word missing).(7) Line 61 - reduce or decrease - I think you meant increase.

Thank you, we have now corrected both sentences.

References

Colloca L, Petrovic P, Wager TD, Ingvar M, Benedetti F. How the number of learning trials affects placebo and nocebo responses. Pain. 2010

Doering BK, Nestoriuc Y, Barsky AJ, Glaesmer H, Brähler E, Rief W. Is somatosensory amplification a risk factor for an increased report of side effects? Reference data from the German general population. J Psychosom Res. 2015